# Long-Term Rewetting of Three Formerly Drained Peatlands Drives Congruent Compositional Changes in Pro- and Eukaryotic Soil Microbiomes through Environmental Filtering

**DOI:** 10.3390/microorganisms8040550

**Published:** 2020-04-10

**Authors:** Micha Weil, Haitao Wang, Mia Bengtsson, Daniel Köhn, Anke Günther, Gerald Jurasinski, John Couwenberg, Wakene Negassa, Dominik Zak, Tim Urich

**Affiliations:** 1Institute of Microbiology, University of Greifswald, Felix-Hausdorff-Str. 8, 17487 Greifswald, Germany; micha.weil@uni-greifswald.de (M.W.); haitao.wang@uni-greifswald.de (H.W.); mia.bengtsson@uni-greifswald.de (M.B.); 2Faculty of Agriculture and Environmental Sciences, University of Rostock, Justus-von-Liebig-Weg 6, 18059 Rostock, Germany; daniel.koehn@uni-rostock.de (D.K.); anke.guenther@uni-rostock.de (A.G.); gerald.jurasinski@uni-rostock.de (G.J.); wakene.chewaka@uni-rostock.de (W.N.); 3Institute of Botany and Landscape Ecology, University of Greifswald, Soldmannstraße 15, 17487 Greifswald, Germany; couw@gmx.net; 4Greifswald Mire Center, Soldmannstraße 15, 17487 Greifswald, Germany; 5Department of Chemical Analytics and Biogeochemistry, Leibniz-Institute of Freshwater Biology and Inland Fisheries, Müggelseedamm 301, 12587 Berlin, Germany; doz@bios.au.dk

**Keywords:** peatland management, soil microbiome, methanogens, sulfate reducers, methanotrophic bacteria, greenhouse gas, methane

## Abstract

Drained peatlands are significant sources of the greenhouse gas (GHG) carbon dioxide. Rewetting is a proven strategy used to protect carbon stocks; however, it can lead to increased emissions of the potent GHG methane. The response to rewetting of soil microbiomes as drivers of these processes is poorly understood, as are the biotic and abiotic factors that control community composition. We analyzed the pro- and eukaryotic microbiomes of three contrasting pairs of minerotrophic fens subject to decade-long drainage and subsequent long-term rewetting. Abiotic soil properties including moisture, dissolved organic matter, methane fluxes, and ecosystem respiration rates were also determined. The composition of the microbiomes was fen-type-specific, but all rewetted sites showed higher abundances of anaerobic taxa compared to drained sites. Based on multi-variate statistics and network analyses, we identified soil moisture as a major driver of community composition. Furthermore, salinity drove the separation between coastal and freshwater fen communities. Methanogens were more than 10-fold more abundant in rewetted than in drained sites, while their abundance was lowest in the coastal fen, likely due to competition with sulfate reducers. The microbiome compositions were reflected in methane fluxes from the sites. Our results shed light on the factors that structure fen microbiomes via environmental filtering.

## 1. Introduction

Peatlands cover 3% of the global land surface and contain about 30% of the global soil organic carbon (SOC) pool in the form of peat [1]. During the agricultural industrialization in the 20th century, many temperate mires in Europe and other parts of the world were drained to be used as grassland or arable land for farming. Globally, drained or generally disturbed peatlands cover 0.3% of the global land surface, but contribute approx. 5% of all anthropogenic greenhouse gas (GHG) emissions including carbon dioxide (CO_2_), methane (CH_4_), and nitrous oxide (N_2_O). The latter also holds for Germany, where drained peatlands cover approx. 5% of the land surface [2,3]. In the German federal state of Mecklenburg–West Pomerania, peatlands cover 13% of the land surface, with their majority being drained (approx. 90%), contributing to about 30% of the total GHG budget in the region [4,5].

Peat accumulates under water-logged conditions, when more plant biomass is formed by growth than is mineralized by the prokaryotic and eukaryotic soil (micro-) biome. Water is a key factor, limiting decomposition by cutting off the supply of oxygen and other alternative terminal electron acceptors (TEA) to mineralizing microbiota [6]. Under anoxic conditions, plant litter accumulates and forms peat with a high content of soil organic matter that accumulates over millennia. Changes in the ecology and hydrology of peatlands can lead to substantial changes in GHG fluxes by changing the peatland biogeochemistry [7]. If peatlands are drained, formerly anoxic soil layers become aerated and carbon is released as CO_2_ during aerobic mineralization by peat microbiota [8].

In the past three decades, several hundred thousand ha of European peatlands were rewetted for environmental protection and to recover some of their ecological functions [9,10]. Long-term rewetting is a proven strategy to protect large SOC stocks; however, it can also lead to increased emissions of the potent GHG CH_4_, and to the release of dissolved organic matter (DOM). While effects of peatland rewetting on GHG and DOM fluxes have been relatively well studied [11,12,13,14,15], the impact on the peat microbiota, the primary driver of GHG production and emissions, is poorly understood, since it has been not well studied in temperate fens (see an exception in Reference [16]). The major players in soil organic matter (SOM) decomposition in peat soils are microorganisms of the bacterial, archaeal, and eukaryotic (e.g., fungi) domains of life, participating in a cascade of SOM degradation steps [9,17] and eventually resulting in emissions of CH_4_ and CO_2_. Hydrolytic extracellular enzymes catalyze the initial decomposition of polymeric SOM. Under anoxic conditions, this step has been considered a major bottleneck and the step determining the rate of downstream processes [8]. Further steps in the degradation are carried out through anaerobic respiratory microorganisms (like denitrifiers, iron reducers, and sulfate reducers) and fermentative and methanogenic microorganisms (Dean et al., 2018), while methanotrophic microorganisms constitute the biological filter for CH_4_ emissions from peat (e.g., Reference [18]). Remarkably, many novel players in soil CH_4_ cycling have been identified in the past few years [19,20,21,22]. Thus, despite more than half a century of research on methanogens, their diversity, physiology, and interactions within the microbiome are still poorly understood. 

It is crucial to obtain a mechanistic understanding of microbial responses and dynamics upon rewetting to suggest robust management practices for climate-optimal peatland protection and utilization (e.g., paludiculture [23]). Environmental filtering, i.e., the selection of certain taxa through abiotic environmental conditions, is considered to be a major mechanism structuring communities [24]. In this context, identifying the main environmental drivers of both total microbiome composition and the abundance of relevant functional groups involved in carbon cycling and methanogenesis is essential.

In this study, we explored the effect of long-term rewetting on temperate fen prokaryotic and eukaryotic microbial communities. For this purpose, the three most relevant fen types in northern Germany were investigated in pairs of drained (dry) and long-term rewetted (wet) fens with contrasting water tables and vegetation: a brackish coastal fen (Coast), a percolation fen (Perco), and an alder carr (Alder). We hypothesized that despite the obvious structural and geo-genetic differences, congruent effects of rewetting would be detectable in the microbiomes of the three peatland types because of environmental filtering. Possible controlling factors of community composition, like DOM quantity and quality, soil moisture, and soil/water salinity, were also determined to allow the effect of rewetting on key players of SOM mineralization and CH_4_-cycling microorganisms, responsible for GHG emissions from peat soils, to be analyzed.

A suite of cultivation-independent microbiological and geochemical methods was applied, including DOM profiling and qPCR of the methanogenesis key enzyme *mcrA* and 16S and 18S rRNA gene-amplicon-based microbiome analysis. Our results may contribute to better understanding of how pro- and eukaryotic microbiomes respond to rewetting and to the relevant controlling environmental factors. This could pave the way for microbiome-based proxies for predicting CH_4_ emissions from rewetted peatlands.

## 2. Materials and Methods

### 2.1. Sites and Sampling

All sampling sites, alder carr, coastal fen, and percolation fen, were located in northeastern Germany (Figure 1). The alder carr is part of the Recknitz river valley, which has been drained since at least 1786 and has been used as a managed forest. The rewetted subsite (Alder_wet_) (54°13′ N 12°49′ E) was unintentionally flooded in 1999 due to a blocked drain and has remained wet since then. In the drained subsite (Alder_dry_) (54°13′ N 12°51′ E), long-term deep drainage has caused subsidence of the soil surface by about 1 m.

The coastal fen, located on a peninsula, is naturally episodically flooded by brackish seawater coming from the Bay of Greifswald of the Baltic Sea, typically during winter storm surges. The entire area was diked in 1850 to allow for pasture and other agricultural use. In 1993, the dykes around the rewetted sub site (Coast_wet_) (54°16′ N 13°39′ E) were removed and, subsequently, the site was episodically flooded again. The drained subsite (Coast_dry_) (54°16′ N 13°39′ E), in contrast, remains behind the dyke and is thus cut off from the seawater influence. Both Coast_wet_ and Coast_dry_ are used as cattle pastures.

The percolation fens, representing the third type of peatlands, are located in the catchment areas of the rivers Trebel and Recknitz. Both sites were deeply drained for intensive land use and were used as deep-drained grassland in the 20th century. In 1998, the water table at the rewetted subsite (Perco_wet_) (54°10′ N 12°74′ E) in the Trebel valley was raised again to 5–10 cm above ground level as part of a large rewetting project (EU LIFE project) (Ministerium für Bau, Landesentwicklung und Umwelt, M-V 1998). In contrast, the water table of the still-drained subsite (Perco_dry_) (54°13′ N 12°63′ E) in the Recknitz valley remains several decimeters below the surface and is mown 1–2 times per year.

At all sites, peat samples were taken in April and August 2017 as triplicate cores at 5–10 cm, 15–20 cm, and 25–30 cm depths (Figure 1) using a gouge. Each sample was homogenized and stored on ice in 50 mL reaction tubes overnight until further processing. The analysis focused on samples taken in April. Samples taken in August were used for qPCR analysis on mcrA gene to verify stable abundance of methanogens.

### 2.2. Soil Physicochemical Properties

Soil moisture was measured gravimetrically by drying 2–3 g of soil over night at 90 °C until mass constancy. Soil moisture was expressed as the percentage of lost water weight to wet soil weight. The potential soil pH was measured at room temperature with a digital pH meter (pH 540 GLP, WTW, Weilheim, Germany) in 0.01 M CaCl_2_ solution with a 1: 2.5 soil to solution ratio. Concentrations of total C, N, and S were measured using a CNS analyzer (Vario MICRO cube—Elementar Analysensysteme GmbH Langenselbold, Germany). DOM was measured in soil extracts prepared by mixing 3 g of previously frozen soil with 30 mL 0.1 M NaCl in 50 mL reaction tubes with subsequent shaking (vortex, 180 rpm) for 30 min. Extracts were filtered through 0.45 µm (pore size) sodium acetate filters, which were prewashed with 50 µL deionized H_2_O to remove soluble acetate. The concentrations and the composition of DOM based on size categories were determined using size-exclusion chromatography (SEC) with organic carbon and organic nitrogen detection (LC-OCD-OND analyzer, DOC-Labor Huber, Karlsruhe, Germany) [25]. The DOM was classified into three major subcategories: (i) “biopolymers”, i.e., non-humic high-molecular-weight substances (>10 kDa) of hydrophilic character and no unsaturated structures like polysaccharides and proteins; (ii) aromatic “humic or humic-like substances” including building blocks; and (iii) “low-molecular-weight substances” including low-molecular-weight acids and low-molecular-weight neutral substances. Fractions were assigned based on standards of the International Humic Substances Society. Detection limits for each fraction were 0.01 mg C L^−1^. Before analysis, all samples were stored at 5 °C for less than two weeks to avoid significant changes of the DOM composition [26]. 

Soluble reactive phosphorus was determined via the ammonium molybdate spectrometric method (DIN EN 1189 D11) using a Cary 1E Spectrophotometer (Varian). N-NH_4_^+^ and N-NO_3_^−^ were determined calorimetrically using the photometry CFA method (Skalar SAN, Skalar Analytical B.V., The Netherlands) according to the guidelines in EN ISO 11732 (DEV-E 23) and EN ISO 13395 (DEV, D 28), respectively.

### 2.3. DNA Extraction and PCR/qPCR

DNA was extracted from 0.25 g soil using the DNeasy PowerSoil Kit (QIAGEN, Hilden, Germany) according to the manufacturer´s recommendation. The bead beating step was performed using a FastPrep^®^-24 5G instrument (MP Biomedicals, Santa Ana, USA), with an intensity of 5.0 for 45 s. DNA concentrations were measured with Qubit^®^2.0 dsDNA High Sensitivity and dsDNA Broad Range assays (Thermo Fisher, Waltham, USA). DNA size was analyzed using 1% agarose gels.

Quantitative polymerase chain reaction was used to measure the abundances of the *mcrA* gene, which codes for methyl coenzyme M reductase subunit A. Quantification of *mcrA* gene copies in peat soil DNA extracts was performed on a qTOWER 2.2 (Analytic Jena, Jena, Germany) using the mlas-mod and mcrA-rev primer pair [27,28]. For each reaction, 15 µL of PCR mixture contained 7.5 µL of innuMIX qPCR MasterMix SyGreen (Analytic Jena, Jena, Germany), 0.75 µL of each primer (10 pmol/µL), 5 µL of ddH_2_O, and 1 µL of template DNA. Duplicate measurements of two concentrations (1 and 2 ng/µL) were performed for each DNA sample. Assay condition was 95 °C for 5 min, 35 cycles of 95 °C for 30 s, 55 °C for 45 s, and 72 °C for 45 °C, followed by melting curve analysis to confirm PCR product specificity. *mcrA* gene copy numbers were calculated from a standard curve obtained by serially diluting a standard from 10^6^ to 10^1^
*mcrA* gene copies per µL. The standard was created with above-mentioned primers by amplifying *mcrA* genes from cow rumen fluid [19] and cloning them into the pGEM^®^-Teasy vector system (Promega, Mannheim, Germany). Amplicons for *mcrA* standard were generated with vector-specific primers sp6 and T7, and the resulting PCR product was cleaned with a DNA purification kit (Biozym, Hessisch Oldendorf, Germany), quality-controlled by agarose gel electrophoresis, and quantified with Qubit^®^2.0 dsDNA High Sensitivity assay. The qPCR assay had the following parameters: slope: 3.41–3.52, efficiency: 0.92–0.96, R^2^: > 0.995.

### 2.4. Amplicon Sequencing and Bioinformatics

16S and 18S rRNA gene amplicons were prepared and sequenced using 300 bp paired-end read Illumina MiSeq V3 sequencing by LGC Genomics (Berlin, Germany). This comprised 16S and 18S rRNA gene PCR amplification, Illumina MiSeq library preparation and sequencing using the primer pair 515YF (GTG YCA GCM GCC GCG GTA A)/B806R (GGA CTA CNV GGG TWT CTA AT) [29] for prokaryotes and 1183F (AATTTGACTCAACRCGGG)/1443R (GRGCATCACAGACCTG) [30] for eukaryotes. The sequences were submitted to European Nucleotide Archive (ENA), Project number PRJEB35436, accession number ERP118476, project name “Changes in peatland microbiome through rewetting”.

The data were processed using R version 3.5.1 [31]. The 16S rRNA and 18S rRNA gene amplicons were denoised using the *dada2* pipeline [32]. The paired forward and reverse sequences were trimmed at 150 bp, and sequences failing to meet the quality check (maxN = 0, maxEE = 2, and truncQ = 2) were discarded. The filtered sequences were then, de-replicated, and clustered into amplicon sequence variants (ASVs) using the *dada2* algorithm with default settings. The paired sequences were merged and chimeric sequences were de-novo checked and removed with *dada2*. The representative sequence of each ASV was then assigned to taxonomy against a modified version of the SILVA SSUref_NR_128 database, containing an updated taxonomy for Archaea [33], using the programs BLASTn [34] and Megan 5 [35]. ASVs that were assigned to chloroplast or mitochondria were removed. Due to the low number of sequences (<500) in one sample (Coast_dry_, core 2, 15–20 cm depth) for both 16S and 18S rRNA genes, this sample was discarded from further analysis. After filtering 1,134,953 and 3,990,919 sequences, sequences were retained and were clustered into 7697 and 5548 ASVs for 16S and 18S rRNA genes, respectively. Plant sequences in 18S rRNA genes, which accounted for 288 ASVs, were filtered for downstream analysis to increase the resolution of other eukaryotic taxa.

Alpha diversity (Shannon entropy) of the microbial community was calculated using the package *phyloseq* [36]. ASV count table was then normalized using metagenomeSeq’s cumulative sum scaling (CSS) [37]. NMDS was performed to analyze the community composition using the package vegan [38] with defaults. Pairwise Spearman’s rank correlations were conducted among soil properties, DNA concentrations, qPCR data, NMDS axes, and methanogen relative abundances using the package *Hmisc* to detect the significance of the impact of the factors on microbial communities [39]. The soil properties with significant impact were fitted into NMDS ordination using the envfit function in the *vegan* package. 

The correlations were visualized by heatmap using *corrplot*. All *p*-values for multiple comparisons were adjusted by the false discovery rate (FDR) method and the null hypothesis was rejected when *p*-values were less than 0.05. 

The microbial functional groups in focus, i.e., fermenters, methanogens, methanotrophs, and sulfate reducers, were predicted from 16S rRNA amplicons using FAPROTAX [40]. Methanogens, methanotrophs, and sulfate reducers were also identified based on literature knowledge; e.g., the ANME2-D group within *Methanosarcina*, known to be methanotrophic using the methanogenic pathway in reverse, was excluded from methanogens. All plots were created using the *ggplot2* package [41].

A co-occurrence network was constructed to explore the potential interactions between species. To eliminate the influence of rare taxa, ASVs with relative abundances lower than 0.05% were discarded. The pairwise Spearman’s rank correlations were calculated with the package *Hmisc* and all *p*-values were adjusted by the FDR. The cut-offs of adjusted *p*-value and correlation coefficient were 0.01 and 0.7, respectively. The network was visualized using the *igraph* package [42]. Topological figures of nodes, including degree (number of a node’s adjacent edges), closeness centrality (the number of shortest paths going through a node), betweenness centrality (number of shortest paths going through a node), and transitivity (probability that the neighbors of a node are connected) were calculated using *igraph*. A network showing the connections between eukaryotes and prokaryotes was subselected from the original network by keeping the edges connecting these two groups. ASVs that were statistically more associated to a certain fen type or water conditions were defined as indicator taxa. The indicators of each group or group combination of a factor (fen type or water condition) were identified using the *indicspecies* package.

### 2.5. Gas Flux Measurements

CH_4_ exchange and ecosystem respiration (CO_2_) from soil and non-woody vegetation were measured in July 2017 with opaque, height-adjustable closed chambers in non-steady-state through-flow mode [43]. Chambers were constructed of flexible polyurethane walls and were connected to circular collars (*n* = 5 per site, 0.63 m diameter) during the measurements. Collars were installed in the soil down to 10 cm depth more than one month prior to measurements. During chamber placement, headspace CO_2_ and CH_4_ concentrations were monitored with portable analyzers (GasScouter^®^, Picarro and Ultraportable Gas Analyzer, Los Gatos Research) [44]. 

## 3. Results

### 3.1. Study Sites and Soil Properties

Our investigations took place in three pairs of drained and rewetted temperate peatlands in northern Germany (see Table 1, Figure 1, and Materials and Methods for details).

In general, the gravimetric water content was higher in the rewetted than in the drained sites (Appendix A). However, comparing the water content mean values of each site (mean of *n* = 9, ± standard deviation (SD)), there was only a minor difference between Perco_dry_ (74.2% ± 5.1) and Perco_wet_ (75.4% ± 5.3), because of a high water table near the soil surface in Perco_dry_ at the time of sampling. In all of the drained sites and in Coast_wet_, the water content increased with depth. In Perco_wet_ and Alder_wet_ (78.7 ± 4.6), with the water table 10 cm above ground, the water content increased slightly with depth or did not change. The lower gravimetric water contents in Coast_dry_ (45.9% ± 6.7), Coast_wet_ (53.8% ± 10.03), and Alder_dry_ (46.3% ± 7.0) corresponded to higher amounts of mineral soil compounds (sand and clay).

The sampling sites were generally acidic to subneutral; pH values varied between fen types but increased with depth in all drained sites. Lower pH values were detected in Coast_dry_ (3.9–4.0) and Alder_dry_ (4.2–4.7), while pH values were higher in Perco_dry_ (5.0–5.2) and Coast_wet_ (4.7–6.1). In Alder_wet_ (5.1–5.1) and Perco_wet_ (5.4–5.4), the pH values did not change with depth (Appendix A).

Salinity in the freshwater fens was generally low (0.08‰–0.49‰). The influence of the nearby brackish bay water with a salinity of 8‰ led to a higher salinity in Coast_dry_ (1.32‰) and in Coast_wet_ (5.2‰) (Appendix A).

DOM concentrations in the soil extracts of Alder_wet_ and Coast_wet_ were higher than in the drained sites. In Perco_wet_, DOM values measured near the surface were higher than in Perco_dry_, but they showed a strong decrease with increasing depth. The distribution of separate DOM fractions, i.e., biopolymers, humic or humic-like substances, and low-molecular-weight organic matter (lCOM) followed the overall pattern of total DOM (Appendix A). 

Total carbon (TC) and total nitrogen (TN) content of peat decreased with depth in Alder_dry_, Coast_dry_, and Coast_wet_. In contrast, TC increased with depth in Alder_wet_, Perco_dry_, and Perco_wet_. In Alder_wet_, Perco_wet_, and Perco_dry_, TN did not change with depth. Total sulfur (TS) content increased with depth in Alder_dry_, Alder_wet_, and Perco_wet_ while it decreased with depth in Coast_dry_ and Coast_wet_. In Perco_dry_, no change was observed over depth (Appendix A).

DNA content per gram dry soil was highest in Perco and Alder_wet_ (> 100 µg g−^1^ DW soil; see Appendix A). In Alder_wet_ and Perco_wet_, DNA content decreased with depth, while in other sites, no distinct trend was observed.

### 3.2. Microbiome Analysis

#### 3.2.1. Diversity of Prokaryotes and Eukaryotes

Alpha diversity of prokaryotes, calculated from 16S rRNA gene amplicon sequence variants (ASVs), was significantly higher in the two freshwater peatlands than in the coastal fen (Appendix A, *p* < 0.001). Alpha diversity was slightly higher in rewetted than in drained sites, but this was significant only in the coastal fen (*p* < 0.001). Alpha diversity showed a positive correlation with soil water content, TC, TN, and DNA content (Appendix A). 

Similarly, alpha diversity of eukaryotes in freshwater peatlands was higher than in the coastal fen (*p* = 0.11). In contrast to prokaryotes, eukaryote alpha diversity was significantly lower in Perco_wet_ than in Perco_dry_ (*p* = 0.015), while no significant difference was observed in Alder and Coast. Alpha diversity of eukaryotes decreased with depth in the rewetted sites. For both pro- and eukaryotes, alpha diversity was negatively correlated with salinity (Appendix A).

#### 3.2.2. Microbiota Composition

Community composition in 16S rRNA and 18S rRNA sequencing analysis of prokaryotes and eukaryotes showed remarkably similar patterns in the non-metric multidimensional scaling (NMDS) plot (Figure 2), with a significant clustering of microbiota from the same sampling sites (PERMANOVA *p* = 0.001). While the communities of freshwater fens Alder and Perco showed some overlap, the microbiota in Coast were clearly separated along NMDS Axis 1, possibly driven by the differences in abiotic factors, such as salinity, total nitrogen (TN), total carbon (TC), C/N ratio, and DOM compounds (Figure 2). Nevertheless, the pro- and eukaryotic microbiota of the dry and wet sites of each fen type were consistently separated along NMDS Axis 2, indicating that the water table of the sites influenced the community composition (PERMANOVA *p* = 0.001, R^2^ = 0.267). Depth showed less impact on community composition (PERMANOVA *p* = 0.161, R^2^ = 0.046), although some clustering according to sample depth was observed.

Pro- and eukaryotic community compositions were compared between fen types, water conditions (dry/wet) and sampling depths (Figure 3a, Figure 4 and Figure A1). We detected a higher relative abundance of Acidobacteria in drained sites (Figure 3a, Appendix A), while Betaproteobacteria, Deltaproteobacteria (mainly comprised of Myxobacteria and sulfate reducers), Chloroflexi, and Bathyarchaeota showed higher values in all rewetted sites and Perco_dry_. SPAM and Latescibacteria were not detected in Coast, but they occurred in the freshwater fens, whereas Actinobacteria showed higher values in Coast than in Alder and Perco. 

The eukaryotic community varied considerably between fen types and water conditions (Figure 3b). Among the fungi, ASVs of Ascomycota (which contains saprophytes, yeasts, and several plant pathogens) showed high relative abundance in each site except Perco_dry_. ASV values of Basidiomycota decreased with depth in Alder and Coast, and showed higher relative abundance in these two sites than in Perco. ASV values of Glomerycota, which form arbuscular mycorrhiza with plants, comprising mostly Mortirellales, were high in Alder_dry_ and Perco_dry_. 

Among the protists, ASV values of Apicomplexa were high (up to 40%) in Perco_dry_, but accounted for less than 5% in other sites. Cercozoa, which contain micro-predators among other lifestyles, had higher values in the freshwater sites Alder and Perco. Relative abundance of Lobosa and Ciliophora, also potential micro-predators, was high in Alder_wet_ and in sites with high water content (rewetted sites and Perco_dry_, respectively). 

Among the Metazoa, Nematoda showed highest relative abundance, followed by Arthropoda. The latter accounted for 5%–10% of total amplicons in Coast and Perco_dry_ and the 5–10 cm layer of Alder_dry_, while their values were lower in Alder_wet_ and Perco_wet_. 

#### 3.2.3. Co-Occurrence Network Analysis

A co-occurrence network was constructed to explore the potential interactions between pro- and eukaryotic ASVs. Indicator ASVs for fen type and water condition were identified. The network showed two distinct clusters of ASVs (Figure 5a), a coastal cluster and a cluster comprised of indicator ASVs from the freshwater sites, highlighting the effect of fen type on the co-occurrence pattern. Many shared indicator ASVs between Perco and Alder (Figure 5a) suggested similarities between these two habitats. In addition, indicator ASVs from the rewetted sites had few connections with those from the drained sites (Figure 5b), highlighting the significant effect of water condition on the co-occurrence pattern. Overall, the co-occurrence network was reminiscent of the individual NMDS plots of pro- and eukaryotes (Figure 2). The degree, closeness centrality, and transitivity of the indicator taxa were higher in the rewetted sites than in the drained sites (Appendix A). These three features were the highest in Coast, followed by Alder and then Perco (Appendix A). The betweenness centrality showed no significant difference.

In the network, the compositions of indicator taxa for fen type reflected the microbiome differences in the three fen types (Figure 3). Acidobcateria, and Betaproteobacteria accounted for the largest proportion in the Alder, while Actinobacteria and Gammaproteobacteria in addition to Acidobacteria dominated the Coast (Figure 5a). In the Perco, Nitrospirae and Alpha- and Betaproteobacteria were the dominant indicator ASVs (Figure 5a). Ascomycota indicator ASVs dominated all three fen types, while Basidiomycota were dominant in the Alder and Coast (Figure 5a). Nematoda contributed to a large proportion of indicator ASVs in the Coast and Perco (Figure 5a). We also identified indicator ASVs for water condition. 

The prokaryotic indicator taxa for the drained sites (Figure 5b) mainly comprised ASVs from Acidobacteria and Alpha- and Gammaproteobacteria, whereas the main indicator taxa of the rewetted sites were Chloroflexi and Beta- and Deltaproteobacteria. Nematoda and Basidiomycota ASVs contributed to a larger proportion in the drained than in the rewetted sites, while Ascomycota showed the opposite pattern (Figure 5b). 

Finally, when exploring the potential interactions between pro- and eukaryotic ASVs, eukaryotes showed higher degrees of connection compared to prokaryotes (Figure 5c). Among the eukaryotes, fungi (Ascomycota, Basidiomycota, and Glomeromycota) and Labyrinthulomycetes possessed the most connections with the prokaryotes.

#### 3.2.4. Functional Groups

##### Methanogens

The abundance of methanogens in the samples, assessed by qPCR of the *mcrA* gene, ranged from 2.1 × 10^5^ to 5.1 × 10^7^ gene copies g^−1^ dry soil (Figure 4). The *mcrA* gene abundance was significantly higher in the three rewetted sites compared with the drained sites (Kruskal–Wallis test, H = 16.5, *p* < 0.001). The difference was largest in the top soils, where *mcrA* abundances were much higher in the rewetted sites. In Alder_wet_ and Coast_wet_, methanogen abundance decreased significantly with depth, while it was rather constant in Perco_wet_. In contrast, methanogen abundance in Alder_dry_ was rather constant or increased with depth (Coast_dry_ and Perco_dry_). Positive correlations were observed between *mcrA* abundance and water, DNA, and DOM content (strongest correlation with biopolymers and lCOM; Appendix A). Furthermore, TC and TN, as well as pH value, were positively correlated with *mcrA* abundance (Appendix A).

The relative abundance of methanogens in the prokaryotic microbiota was low. Methanogen-affiliated ASVs in the amplicon sequence datasets were on average below 0.25% of total 16S rRNA amplicons (Figure A1). In most of the drained fens, methanogen ASVs were even absent, although they were detectable by *mcrA* qPCR (Figure 4). The upper soils of Alder_wet_ and Coast_wet_ showed the highest abundances of methanogens, while the methanogens were more abundant in the subsoils of Perco_wet_. Methanobacteriales and Methanosarcinales (including the families Methanosaetaceae and Methanosarcinaceae) were the dominant orders in Alder_wet_ and Coast_wet_, while methanogens were more diverse in Perco_wet_, where Methanomassiliicoccales and Methanomicrobiales were also present.

Despite apparently missing out on some methanogens in the amplicon data, the relative abundance calculated from *mcrA* gene copies per g dry soil (qPCR) showed a clear positive correlation with the relative abundance of methanogenic archaea in 16S rRNA amplicon (Spearman’s rank correlation, rho = 0.578; *p* < 0.001).

##### Methanotrophs

16S rRNA genes of known methanotrophic prokaryotes as assigned via FAPROTAX analysis, were found in all sites (Figure 6), but never exceeded 1% of all prokaryotic 16S rRNA genes on average. Their relative abundance and taxon composition differed both between fen types and dry/wet. In drained fens ASVs affiliated with Ca. Methylacidiphilum (phylum Verrucomicrobia) were dominant across the soil profile (Figure A1). In contrast, ASVs of Methylococcaceae (class Gammaproteobacteria) were the most abundant methanotrophs in Coast_wet_ and Perco_wet_. Putative anaerobic methanotrophs affiliated with Ca. Methanoperedens (ANME-2d, archaea) and with Ca. Methylomirabilis (NC10 phylum) were the dominant methanotroph guild at the Alder_wet_ site.

##### Sulfate Reducers

ASVs assigned as sulfate reducers via FAPROTAX analysis were generally more abundant than methanogens and methanotrophs (between 0.1% and 5% on average, Figure 6). Like methanogens, their relative abundance was higher in rewetted sites than in drained sites. Highest relative abundance was observed in the Coast_wet_, which is exposed to recurrent influx of sea water. ASVs of Desulfobacterales were particularly abundant in this site (Figure A1). Desulfarculales and Desulfosporosinus were only detected in rewetted sites, while Desulfuromonadales occurred in all sites. Relative abundance of sulfate reducers was positively correlated to salinity, but not to total sulfur (Appendix A).

### 3.3. Greenhouse Gas Emissions

To test predictions from the microbiome analysis, GHG fluxes on the sites were measured at a later time-point (July 2017), in addition to obtaining additional *mcrA* qPCR data. In all three drained peatlands, a net uptake of CH_4_ was observed (Table 2). In contrast, net emissions of CH_4_ were observed in two of the rewetted sites, with the highest CH_4_ emission of 15.8 mg m^−2^ h^−1^ CH_4_ found for Perco_wet_. Similarly, *mcr*A genes were more abundant in rewetted sites than in drained, with the highest *mcr*A copy numbers in Perco_wet_. The *mcrA* gene abundances did not differ much between the two time-points (Table 2), indicating a rather stable-sized methanogen community. Ecosystem respiration was highest in coastal peatlands. In all sites, the drained peat soil emitted higher amounts of CO_2_. 

## 4. Discussion

To our knowledge, this is the first comprehensive study investigating the impact of long-term rewetting (i.e. more than 10 years) on pro- and eukaryotic freshwater and coastal fen microbiomes. The parallel study of three different fen types allowed for the identification of overarching factors alongside fen-type-specific factors. The NMDS plots of both prokaryotes and eukaryotes were almost superimposable. It is remarkable that rewetting caused highly congruent effects on the beta-diversity of all three domains of life, and it is conceivable that water table and salinity were the main environmental filters among the physicochemical controlling factors. Long-term rewetting resulted in more anoxic conditions in the rewetted sites compared with the drained sites. Increased anoxia is supported by several lines of evidence, including (1) a higher moisture content, (2) higher proportions of presumably fermentative microorganisms, and (3) higher proportions of taxa with different types of anaerobic respiration, such as methanogenesis and sulfate reduction. 

### 4.1. Environmental Filtering Effects on Pro- and Eukaryotic Community Composition 

The co-occurrence network analysis showed that more taxa indicative for each of the different fen types than for water status were detected, which agreed well with the community composition analysis: fen type had a more significant effect on the community assemblages than water condition. 

The higher degree, closeness centrality, and transitivity of indicator ASVs in the rewetted sites suggested more connections in the microbiomes. This might have been due to the fact that higher water flow rates in soils create greater homogeneity and hence weak niche differentiation, contributing to stronger interactions between soil microbes. Furthermore, it might reflect lower oxygen availability leading to close metabolic interactions in the anaerobic microbiomes required for plant biomass degradation [18,46]. In fact, the main indicator ASVs in the rewetted sites were Chloroflexi and Delta- and Betaproteobacteria, which contain microbes known from anoxic environments [47]. In the drained sites, most indicator ASVs were Acidobacteria, the predominant phylum, likely reflecting the general acidic to subneutral pH conditions. Drainage and subsequent oxidation processes in peat soils are associated with acid production, depending on the buffer capacity of the soil. The pH may drop by one or two units compared with pre-drained conditions [48]. In contrast, rewetting with base-rich water will increase alkalinity and pH [49]. In particular, the two dominant classes, Acidobacteria (Acidobacteria Subdivision 1) and Solibacteres (Acidobacteria Subdivision 3), decreased in relative abundance upon rewetting (Appendix A), which is in line with the obligatory aerobic lifestyle of most characterized species [50]. Members of the aerobic Acidobacteria play important roles in carbohydrate degradation, but some subdivisions also contain facultative or obligate anaerobic species [50,51]. Other putative anaerobic phyla such as Ignavibacter and Bathyarchaeota occurred only in the rewetted sites and deep layers of drained sites. Next to prokaryotes, water level impacted also the eukaryotes. For instance, Nematoda and Basidiomycota indicator ASVs accounted for a larger proportion in drained sites, likely because they are dependent on oxic conditions due to their aerobic energy metabolism. Obviously, increased soil moisture in long-term rewetting acted as an environmental filter leading to community compositional changes from oxygen-dependent taxa to anaerobic and fermentative taxa (Figure 6).

A second environmental filter in our study was sea water intrusion. Higher salinity in the Coast resulted in the lowest diversity of both prokaryotes and eukaryotes (Appendix A). This might have led to a weaker niche differentiation as compared with the freshwater sites. In consequence, it may have contributed to stronger connections, observed as the higher connectivity and transitivity of indicators from Coast. This environmental filtering through intrusion of sea water in these sites also led to a change in microbial community with increases of sulfate reducers and decrease of methanogens ([52] and see below).

In the co-occurrence network, we found fewer eukaryotic than prokaryotic ASVs, suggesting that one eukaryote was probably connected with multiple prokaryotes, as implied by higher degrees of connection for eukaryotes. Similarly, the richness of eukaryotes was far less than the richness of prokaryotes, which is common for soil microbiota, as shown before in, for example, Reference [18]. Therefore, our results potentially suggest this as a universal rule of pro-eukaryote interactions. Remarkably, the prokaryotes were mostly connected with fungi like Ascomycota, Basidiomycota and Glomeromycota, suggesting that fungi may drive the composition of prokaryotic community. 

### 4.2. The Influence of Rewetting on Methanogens

The important group of methanogenic archaea was hardly detected in the analysis of microbial networks and played no role in the determination of indicator taxa. The reason was the low relative abundance of methanogens in the amplicon datasets of the microbiota, being less than three per million in all samples. Although their relative abundance in the microbiota was low even under rewetted conditions, their abundance per gram soil was high, especially in the rewetted sites. Wet peatlands are known to contribute a large share of the global CH_4_ emissions [1,53]. The higher *mcrA* abundance after rewetting (Figure 4) could lead to increased CH_4_ emissions from peatlands, as is the case in rice paddy fields [16,54]. Interestingly, their abundance was high in the rewetted top soils—at least in Alder_wet_ and Perco_wet_—and not in the subsoils. This is in contrast to general assumptions about their vertical distribution in the soil profile (reviewed in Reference [6]) and corroborates recent findings from other ecosystems [55,56]. This observation makes peatland management practices such as top soil removal attractive for mitigation of CH_4_ emissions from peatlands [57,58]. The abundance of methanogens was strongly correlated with soil water content (Appendix A), which drives anoxia in the soil. However, many other factors, like substrate availability of DOM, especially lCOM, C, and N content of soil, degradation status of the peat, and sulfate and nitrate concentration all positively correlated with methanogen abundance per gram dry soil as well (Appendix A). One significant factor was salinity, resulting from brackish water intrusions from the Baltic Sea in Coast_wet_, which led to large differences between freshwater fens (Alder and Perco) and Coast. In Coast_wet_, the abundance of methanogens per gram soil was at least 10-fold lower than in the freshwater sites Alder_wet_ and Perco_wet_. Thus, this environmental filtering through salinity likely leads to lower CH_4_ emissions, as also shown before in several studies [47,59,60]. 

The strong positive correlation of *mcrA* gene abundance in qPCR with the relative abundance of methanogenic archaea in 16S rRNA gene amplicon datasets confirmed the validity of the measured results. However, in most of the drained sites (Perco_dry_, most Alder_dry_ and Coast_dry_), methanogenic taxa were not present in the dataset, although *mcrA* genes were quantified by qPCR. This can be explained by the rather shallow average sequencing depth of 20,000 reads per sample; one single read would correspond to 0.05‰ in the microbiota, and thus taxa with lower abundances could not be detected. Still, the distribution of methanogenic taxa led to the assumption that substrate availability was different in the sites. While hydrogenotrophic Methanobacteriales were dominant in Coast_wet_, Coast_dry_, and Alder_wet_, Perco_wet_ showed a high variability in methanogenic groups. Here, the Methanosarcinales families of Methanosaetaceae (obligate acetoclastic) and Methanosarcinaceae (flexible substrate spectrum) occurred, as did the Methanomicrobiales and the Methanomassiliicoccales (which use the H_2_ dependent methylotrophic pathway). Methanomassiliicoccales were detected in every site with Methanomassiliicoccales-specific qPCR primers AS1/AS2 [61] (data not shown). Their high abundance in Perco_wet_ indicates that besides the well reported hydrogenotrophic and acetotrophic pathways [15], methylotrophic methanogenesis may contribute more to peatland CH_4_ emissions than currently assumed [19,62].

### 4.3. Interactions between Methanogens, Methanotrophs, and Sulfate Reducers

The second important player in peatland CH_4_ emissions is the heterogenic group of methanotrophic Bacteria and Archaea, acting as biological CH_4_ filter through aerobic or anaerobic oxidation of the produced CH_4_ (reviewed in Reference [53,63]). Do the methanogens influence abundance of methanotrophs? The methanotrophic groups differed with fen type, and were generally lower abundant in the drained sites where Ca. Methylacidiphilum aerobic CH_4_ oxidizers were dominant. These organisms were originally isolated from acidic and volcanic environments, but have recently also been found in pH neutral environments [64,65]. They are characterized by growth at low CH_4_ concentrations, and are possibly atmospheric CH_4_ oxidizers, which would make them independent of substrate supply by methanogens. As found in other wetland studies (e.g., Reference [66]), aerobic CH_4_ oxidizers of Methylococcaceae dominated in the wet sites. The anaerobic group Ca. Methylomirabilis [65], which uses oxygen from nitrite to reduce CH_4_, appeared in the sites with the highest water content (Alder_wet_, Perco_dry_, and Perco_wet_). The anaerobic CH_4_-oxidizing ANME-2d (Ca. Methanoperedens), which reduces CH_4_ by the reverse methanogenic pathway [20], were detected in the water-covered fen Alder_wet_. Although we did not find a clear correlation between methanogen and methanotroph abundance (see Reference [16] for similar findings), we did find a strong distinction of anaerobic and aerobic methanotrophs depending on water content, and therefore oxygen availability, between sites and depths.

What is the basis of the environmental filter leading to the low methanogen abundance in the coastal fen? It is known that sulfate reducers can have an impact on methanogens, because methanogens and sulfate reducers not only share the anaerobic niche, but also compete for the same substrates, such as hydrogen [47,67,68] and acetate [69]. Due to their higher substrate affinity, sulfate reducers can outcompete methanogens, leading to lower methanogen abundance in the presence of sulfate reducers [60,69]. In fact, Desulfobacterales, known competitors for H_2_ as a substrate for methanogenesis, occurred in the coastal site Coast_wet_ at a large proportion, which was associated with a more than 10-fold lower *mcrA* gene abundance than in Alder_wet_ and Perco_wet_. The influence of salinity in tidal marshes is also known to decrease CH_4_ emission compared to freshwater sites [60,70]. 

These inferences based on microbial analysis and soil parameters were confirmed by GHG flux measurements, conducted three months after microbial sampling. For instance, ecosystem respiration in the drained sites was significantly higher than in the wet sites, in line with the higher abundance of aerobic pro- and eukaryotes that presumably oxidize more soil organic carbon. As suggested by the higher abundance of methanogens (Table 2 and Figure 4), CH_4_ emissions were higher in wet sites compared to the drained sites. Interestingly, all drained sites as well as Coast_wet_ were CH_4_ sinks in the summer, presumably fostered by the activity of the methanotrophs. In contrast, the wet sites Alder_wet_ and Perco_wet_, with mcrA abundances < 10^7^ g^−1^ dry weight at the time of sampling, acted as CH_4_ sources. 

## 5. Conclusions

In conclusion, this study provides microbiome-based insights into the environmental filtering effects of rewetting of three fen ecosystems, alder carr, percolation fen, and coastal fen. Since the practice of rewetting previously drained mires is gaining more attention as a measure to mitigate GHG emissions from soils, our study provides key knowledge about the effects on the resident microbiomes. In all systems, soil moisture was the driving factor for change in pro- and eukaryotic microbial community and GHG fluxes. In the coastal fen, the inflow of brackish water seemed to stimulate a high abundance of sulfate reducers, suppressing methanogen abundance and methanogenic activity. This suggests that, in terms of minimizing CH_4_ emissions after rewetting, increased sulfate content and/or topsoil removal might be promising options. Future analysis of seasonal microbiome dynamics and measurements of CH_4_ fluxes will provide evidence as to whether these predictions will hold true.

## Figures and Tables

**Figure 1 microorganisms-08-00550-f001:**
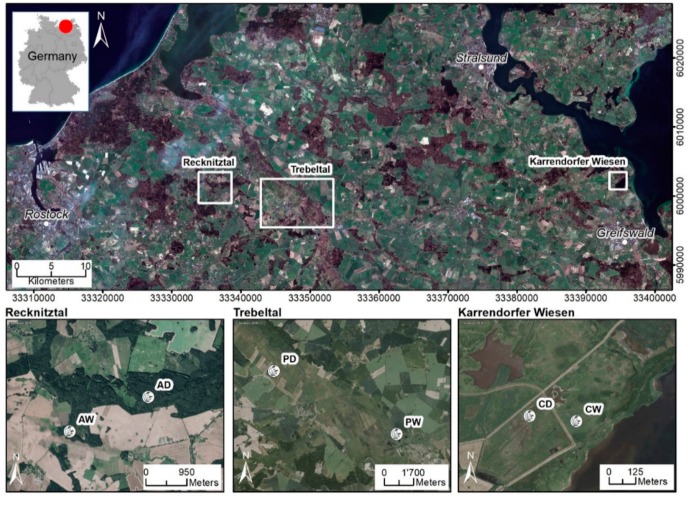
Overview of the six sampling sites: Alder_dry_ (AD), Alder_wet_ (AW), Coast_dry_ (CD), Coast_wet_ (CW), Perco_dry_ (PD), and Perco_wet_ (PW).

**Figure 2 microorganisms-08-00550-f002:**
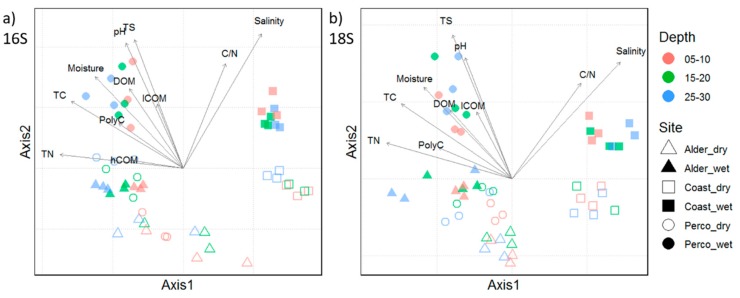
NMDS plots showing (**a**) prokaryotic and (**b**) eukaryotic community compositions. Symbols show triplicates of cores and indicate the study sites; the color indicates the depth in cm (see legend). Environmental vectors with a significance of *p* < 0.05 are shown in black. TN: total nitrogen, hCOM: humic like carbon, TC: total carbon, PolyC: carbon polymers, DOM: dissolved organic matter, lCOM: low-molecular-weight organic matter, TS: total sulfur, C/N: carbon/nitrogen ratio. Triangles: alder, squares: coast, circles: Perco. Open symbols: dry, filled symbols: wet.

**Figure 3 microorganisms-08-00550-f003:**
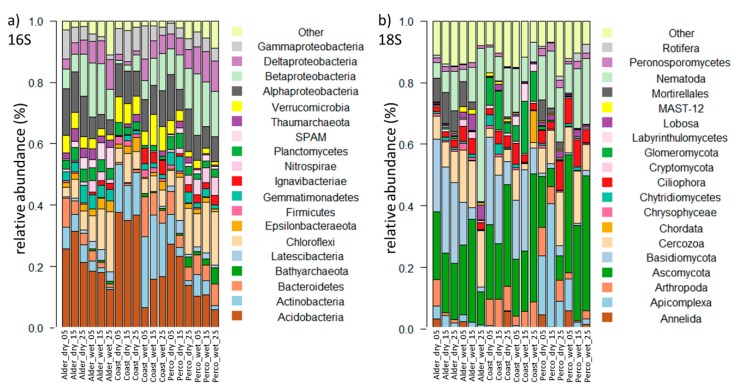
Composition of prokaryotic and eukaryotic microbial communities of drained and rewetted fens. (**a**) Taxonomic composition of prokaryotic 16S rRNA genes displayed at the phylum level, with Proteobacteria shown at the class level. (**b**) Taxonomic composition of eukaryotic 18S rRNA genes, at the phylum level. Bars show triplicates of cores; phyla with <1% abundance are displayed as “other”. The sample ID on the x-axis displays fen type and depth in cm.

**Figure 4 microorganisms-08-00550-f004:**
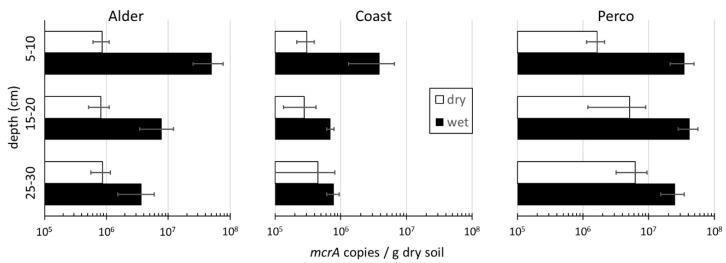
Abundance of methanogens. mcrA gene copy numbers per gram of dry soil based on qPCR. X-axis shows mcrA copies per g dry soil: drained sites in white, rewetted sites in black bars, and error bars show standard deviation of triplicate soil cores.

**Figure 5 microorganisms-08-00550-f005:**
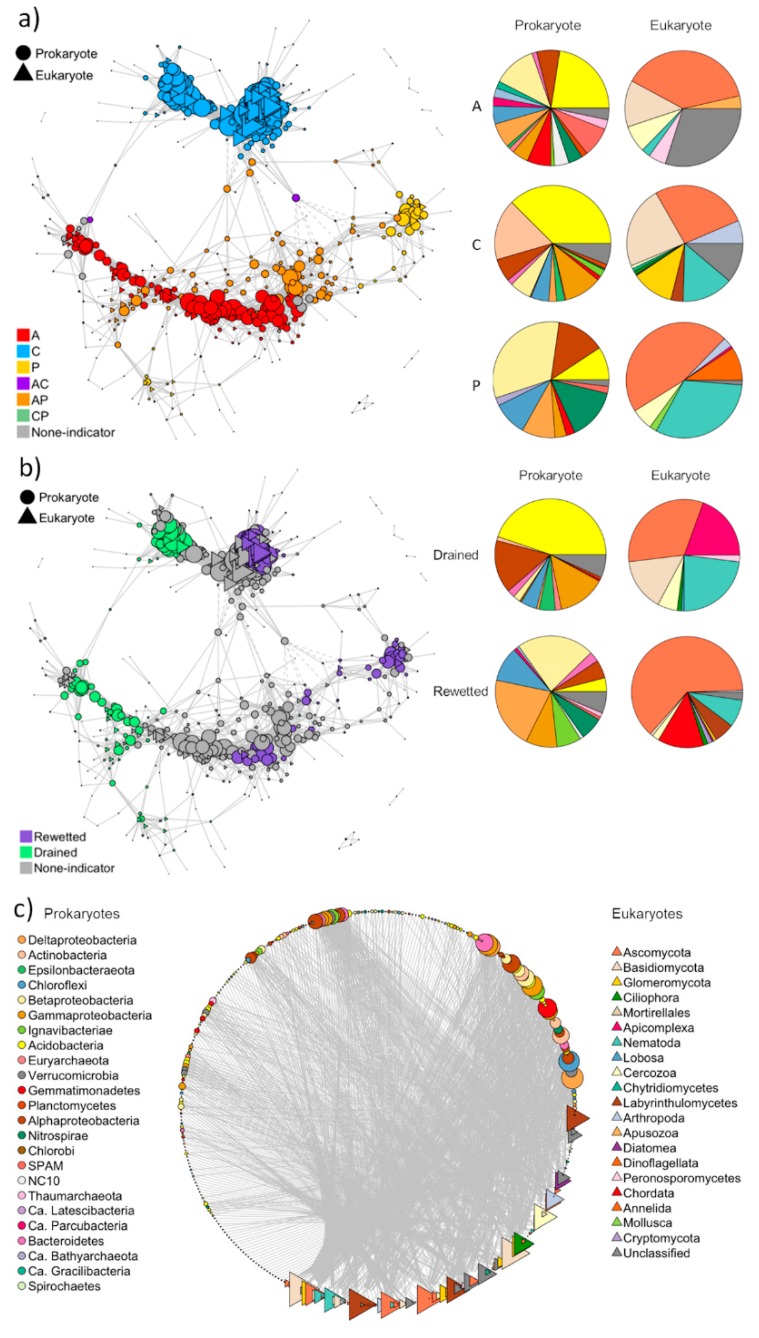
Co-occurrence networks. (**a**) Co-occurrence network of indicator ASVs associated with (A) alder carr, (C) coastal fen, and (P) percolation fen, or with both alder carr and coastal fen (AC), both alder carr and percolation fen (AP), and both coastal fen and percolation fen (CP). (**b**) Co-occurrence network of indicator ASVs associated with drained and rewetted sites. (**c**) Co-occurrence network showing connections between prokaryotic and eukaryotic ASVs. ASVs with relative abundances lower than 0.05% were discarded. ASVs were identified as indicators associated with fen type or water condition using the *indicspecies* package [45], and are color-coded according to fen type and water condition (drained/rewetted) in (**a**) and (**b**), respectively. The relative abundances of prokaryotes and eukaryotes associated with the different fen types and water conditions are shown as pie charts on the right side of (**a**) and (**b**), respectively. The color code in pie charts of (**a**) and (**b**) correspond to the color code in (**c**). Solid lines indicate significant positive Spearman’s rank correlations (R > 0.7. *p* < 0.01), while dashed lines indicate significant negative correlations (R < −0.7. *p* < 0.01). Nodes are sized according to their degrees.

**Figure 6 microorganisms-08-00550-f006:**
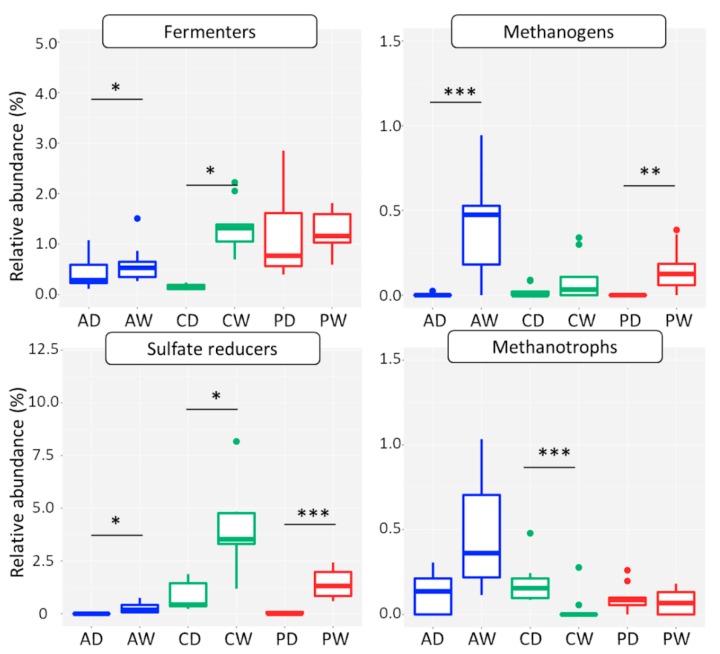
Relative abundance of functional groups in 16S rRNA gene amplicon datasets. Boxplots show relative abundance in depth and triplicate samples. X-axes represent six sites: Alder_dry_ (AD), Alder_wet_ (AW), Coast_dry_ (CD), Coast_wet_ (CW), Perco_dry_ (PD), and Perco_wet_ (PW). Functional groups were assigned from 16S rRNA gene amplicon data by FAPROTAX analysis. Asterisks indicate the significant difference between the drained and rewetted sites (Kruskal-Wallis test, * *p* < 0.05, ** *p* < 0.01, *** *p* < 0.001).

**Table 1 microorganisms-08-00550-t001:** Sampling site characteristics.

Abbr.	Fen Type	State (Year of Rewetting)	Water Table	Peat Thickness	Dominant Plant Species
Alder	Alder carr	dry	−100 cm	60 cm	Black Alder (*Alnus glutinosa)*, Ground Elder (*Aegopodium podagraria*), and Common Nettle (*Urtica dioica*)
		wet (1999)	+15 cm	>1 m	Black Alder (*Alnus glutinosa*), Greater Pond Sedge *(Carex riparia)*
Coast	Coastal fen	dry	−70 cm	70 cm	Creeping Bentgrass (*Agrostis stolonifera*)
		wet (1993)	−5 cm	30 cm	Creeping Bentgrass (*Agrostis stolonifera*)
Perco	Percolation fen	dry	−20 cm	6 m	Creeping Buttercup (*Ranunculus repens*), Tufted Hairgrass (*Deschampsia cespitosa*)
		wet (1998)	+10 cm	6 m	Lesser Pond Sedge (*Carex acutiformis*)

**Table 2 microorganisms-08-00550-t002:** Fluxes of CH_4_ and ecosystem respiration (measured as CO_2_ flux) on all study sites compared to methanogen abundance (mcrA_DW). Negative flux values represent uptake from the atmosphere. Table shows mean values (*n* = 5 for CH_4_ and ecosystem respiration CO_2_ fluxes, *n* = 9 for mcrA_DW) and standard deviation. Significance was tested with Kruskal–Wallis test.

Site	CH_4_ Flux (mg m^−2^ h^−1^)		Ecosystem Respiration CO_2_ Flux (mg m^−2^ h^−1^)		mcrA_DW (Copies g^−1^ Soil Dry Weight) April 2017		mcrA_DW (Copies g^−1^ Soil Dry Weight) August 2017	
Alder_dry	−0.08	± 0.07	493	± 406	8.5 × 10^5^	± 2.9 × 10^5^	3.14 × 10^5^	±3.30 × 10^4^
Alder_wet	0.88 *	± 0.51	47 ^†^	± 70	2.1 × 10^7^	± 2.6 × 10^7^	3.89 × 10^7^	±1.66 × 10^6^
Coast_dry	−0.11	± 0.14	2024	± 585	3.5 × 10^5^	± 2.5 × 10^5^	3.62 × 10^5^	±9.97 × 10^4^
Coast_wet	0.00 *	± 0.01	1420	± 311	1.8 × 10^6^	± 2.1 × 10^6^	5.32 × 10^5^	±6.90 × 10^4^
Perco_dry	−0.12	± 0.05	783	± 152	4.3 × 10^6^	± 3.5 × 10^6^	6.31 × 10^6^	±1.07 × 10^6^
Perco_wet	15.8 *	± 23.8	550 ^†^	± 232	3.4 × 10^7^	± 1.5 × 10^7^	5.28 × 10^7^	±3.55 × 10^6^

* significantly higher than in drained site (*p* < 0.05). ^†^ significantly lower than in drained site (*p* < 0.05).

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
