# Peer review of "Long-Term Rewetting of Three Formerly Drained Peatlands Drives Congruent Compositional Changes in Pro- and Eukaryotic Soil Microbiomes through Environmental Filtering"

_microorganisms, 2020, doi:10.3390/microorganisms8040550_

Round 1

Reviewer 1 Report

General comment

In this study, the authors investigated the effect of peatland rewetting on prokaryotic and eukaryotic community structure/composition and variation using culture-independent 16S rRNA and 18S rRNA amplicon sequencing methodology. Furthermore, soil properties were estimated using DOM analyses. Quantitative PCR of methanogenesis enzyme-coding gene mcrA revealed the abundance of methanogens in topsoils after rewetting which has a positive correlation with the salinity. Authors put forward the soil management strategies such as topsoil removal and/or increasing the salinity.

I liked the hypothesis and reading the manuscript. It is clear and concise. I do not have any major comments. Please see the following minor comments.

Introduction

Line 46-47 – “when more plant biomass is formed by growth than is mineralized by the prokaryotic and eukaryotic soil (micro-)biome”. Is it “growth, then”?

Line 94 – Please revise it “Eventually”

Lines 99-10 – “was drained at least from 1786 on” or “was drained at least from 1786 onwards”

Materials and methods

Lines 157-158 – Primer “mlas-mod” is not mentioned in the reference 27 (Steinberg, L.M.; Regan, J.M. mcrA-targeted real-time quantitative PCR method to examine methanogen communities. Appl. Environ. Microbiol) please clarify it

Also, rephrase the sentence. “was performed on a qTOWER 2.2 (Analytic Jena, Jena, Germany) using mlas-mod and mcrA- rev primer pairs [27]

Line 180- Please rephrase to “gene amplicons”

Line 180-182 – Was DAADA used with default parameters?

Line 184 – Change to “database”

Line 186 – Change from “filtered to removed”

Table 1- Use the normal font for  “Greater Pond Sedge

Figure S1 – Would be helpful if authors mention the code abbreviations for different peatlands

Line 292- Shouldn’t’t Figure A1 be a supplementary figure?

Lines 523-524 – It is detected that the abundance of methanogens and methanotrophs is not correlated. Out of interest, are there any other microbes detected (could be in low abundance) in this study that can utilize the methane?

Table S1- Please remove extra blank page

Author Response

Answers to reviewer 1

In this study, the authors investigated the effect of peatland rewetting on prokaryotic and eukaryotic community structure/composition and variation using culture-independent 16S rRNA and 18S rRNA amplicon sequencing methodology. Furthermore, soil properties were estimated using DOM analyses. Quantitative PCR of methanogenesis enzyme-coding gene mcrA revealed the abundance of methanogens in top soils after rewetting which has a positive correlation with the salinity. Authors put forward the soil management strategies such as topsoil removal and/or increasing the salinity.

I liked the hypothesis and reading the manuscript. It is clear and concise. I do not have any major comments.

We are happy that the reviewer appreciates the quality of our study.

Please see the following minor comments.

Introduction

Line 46-47 – “when more plant biomass is formed by growth than is mineralized by the prokaryotic and eukaryotic soil (micro-)biome”. Is it “growth, then”?

It is “growth than”. More plant mass grows than is mineralized. The Plant material that is not mineralized builds up the peat.

“Peat accumulates under water-logged conditions, when more plant biomass is formed by growth than is mineralized by the prokaryotic and eukaryotic soil (micro-)biome.” (Line 46-47)

Line 94 – Please revise it “Eventually”

It was changed according to the reviewer’s suggestion. (Line 93)

Lines 99-100 – “was drained at least from 1786 on” or “was drained at least from 1786 onwards”

It was changed according to the reviewer’s suggestion. (Line 100)

Materials and methods

Lines 157-158 – Primer “mlas-mod” is not mentioned in the reference 27 (Steinberg, L.M.; Regan, J.M. mcrA-targeted real-time quantitative PCR method to examine methanogen communities. Appl. Environ. Microbiol) please clarify it

We thank reviewer 1 for this comment. An additional reference was added.

(Angel, R.; Matthies, D.; Conrad, R.; Meyer, A.; Kilian, E. Activation of Methanogenesis in Arid Biological Soil Crusts Despite the Presence of Oxygen. PLoS One 2011, 6, e20453.)

Also, rephrase the sentence. “was performed on a qTOWER 2.2 (Analytic Jena, Jena, Germany) using mlas-mod and mcrA- rev primer pairs [27]

The sentence was rephrased as suggested

“Quantification of mcrA gene copies in peat soil DNA extracts was performed on a qTOWER 2.2 (Analytic Jena, Jena, Germany) using the mlas-mod and mcrA-rev primer pair [27,28].” (Line 158-160)

Line 180- Please rephrase to “gene amplicons”

Sentence was rephrased. Please refer to the next comment.

Line 180-182 – Was DAADA used with default parameters?

We followed the standard tutorial from the dada2 official website (https://benjjneb.github.io/dada2/tutorial_1_8.html), in which most steps were done with default settings. To make it clearer, we provide more details about this part in the revised manuscript.

“The data were processed using R version 3.5.1 [31]. 16S rRNA and 18S rRNA gene amplicons were denoised using dada2 pipeline [32]. The paired forward and reverse sequences were trimmed at 150 bp, and sequences failing to meet the quality check (maxN=0, maxEE=2, and truncQ=2) were discarded. The filtered sequences were then, de-replicated and clustered into amplicon sequence variants (ASVs) using dada2 algorithm with default settings. The paired sequences were merged and chimeric sequences were de-novo checked and removed with dada2.” (Line 182-187).

Line 184 – Change to “database”

Changed as suggested. (Line 189)

Line 186 – Change from “filtered” to “removed”

Changed as suggested. (Line 191)

Table 1- Use the normal font for “Greater Pond Sedge

Font was changed to normal font.

Figure S1 – Would be helpful if authors mention the code abbreviations for different peatlands

Code is already mentioned in the figure legend of S1.

Line 292- Shouldn’t Figure A1 be a supplementary figure?

We put it as an appendix figure instead of a supplementary figure to have an easy access for the reader of the main manuscript.

Lines 523-524 – It is detected that the abundance of methanogens and methanotrophs is not correlated. Out of interest, are there any other microbes detected (could be in low abundance) in this study that can utilize the methane? 

We did not find other potential methanotrophs that could utilize the methane. However, not all methanotrophic microorganisms are currently known by 16S rRNA genes, e.g. USC-gamma. Thus, a positive correlation between both functional groups might well exist.

Table S1- Please remove extra blank page

Extra page was removed.

Reviewer 2 Report

The work concerns an important topic - the impact of human activity on greenhouse gases emissions from peat bog areas undergoing renaturalization. It should be noted that the authors have made a significant contribution to the preparation of the article, providing a versatile dataset and subjecting the obtained results to an in-depth statistical analysis. I suppose that application of qPCR was a means to solve the problem that authors have encountered with detection of the methanogenic Archaea via NGS. Higher sequencing depth could also help but not necessarily. Archaea are usually low in counts and sometimes even several times higher number of reads does not necessarily lead to their detection. For future studies I would advise the authors to merge DNA from 2-3 isolates as it was proven to increase detection of rare taxa (e.g. methanogens). 

Nevertheless, the article is written well and it's very informative. It require language editing by native speaker as there are repetitions and some sentencess are confusing;

e.g. line 327-329; 353-354; 376-377

line 497-498 how was taxon detection limit determined? It needs to be explained.

Conclusions section - needs to be improved to correspond with the goals set at the beginning of the article (lines 84-89)

Furthermore, conclusion that increase in salinity may be a means to overcome methane emissions it a bit controversial. Increased salinity is a problem itself so authors suggest to solve a problem by creating another problem. 

In spite those remarks I find the paper worth publishing. 

Author Response

Answers to reviewer 2

The work concerns an important topic - the impact of human activity on greenhouse gases emissions from peat bog areas undergoing renaturalisation. It should be noted that the authors have made a significant contribution to the preparation of the article, providing a versatile dataset and subjecting the obtained results to an in-depth statistical analysis.

We are happy that the reviewer appreciates the quality of our study.

I suppose that application of qPCR was a means to solve the problem that authors have encountered with detection of the methanogenic Archaea via NGS. Higher sequencing depth could also help but not necessarily. Archaea are usually low in counts and sometimes even several times higher number of reads does not necessarily lead to their detection. For future studies I would advise the authors to merge DNA from 2-3 isolates as it was proven to increase detection of rare taxa (e.g. methanogens). 

The usage of methanogen-specific qPCR enabled their quantification per gram soil, as opposed to relative abundance data derived from NGS amplicon data sets.

Nevertheless, the article is written well and it's very informative. It requires language editing by native speaker as there are repetitions and some sentences are confusing;

e.g. line 327-329; 353-354; 376-377

As suggested, we have extensively edited the language to improve readability. We also specifically focused on the mentioned section. It reads now as follows:

“3.2.3. Co-occurrence network analysis

A co-occurrence network was constructed to explore the potential interactions between pro- and eukaryotic ASVs. Further, indicator ASVs for fen type and water condition were identified. The network showed two distinct clusters of ASVs (Figure 4a) a coastal cluster and a cluster comprised of indicator ASVs from the freshwater sites, highlighting the effect of fen type on the co-occurrence pattern. Many shared indicator ASVs between Perco and Alder (Figure 4a) suggested similarities between these two habitats. In addition, indicator ASVs from the rewetted sites had few connections with those from the drained sites (Figure 4b), highlighting the significant effect of water conditions on the co-occurrence pattern. Overall, the co-occurrence network is reminiscent of the individual NMDS plots of pro- and eukaryotes (Figure 2). The degree, closeness centrality and transitivity of the indicator taxa were higher in the rewetted sites than in the drained sites (Figures S4). These three features were the highest in Coast, followed by Alder and then Perco (Figure S4). The betweenness centrality showed no significant difference. (Line 327-339)

In the network, the compositions of indicator taxa for fen type reflected the microbiome differences in the three fen types (Fig. 3). Acidobcateria, and Beta-Proteobacteria accounted for the largest proportion in the Alder, while Actinobacteria and Gamma-Proteobacteria in addition to Acidobacteria dominated the Coast (Figure 4a). In the Perco, Nitrospirae, Alpha- and Beta-Proteobacteria were the dominant indicator ASVs (Figure 4a). Ascomycota indicator ASVs dominated all three fen types, while Basidiomycota were dominant in the Alder and Coast (Figure 4a). Nematoda contributed to a large proportion of indicator ASVs in the Coast and Perco (Figure 4a). We also identified indicator ASVs for hydrological status.

The prokaryotic indicator taxa for the drained sites (Figure 4b) mainly comprised ASVs from Acidobacteria and Alpha- and Gamma-Proteobacteria, whereas the main indicator taxa of the rewetted sites were Chloroflexi, Beta- and Delta-Proteobacteria. Nematoda ASVs contributed to a larger proportion in the drained than in the rewetted sites while Ascomycota showed the opposite pattern (Figure 4b).

Finally, when exploring the potential interactions between pro- and eukaryotic ASVs, eukaryotes showed higher degrees of connection compared to prokaryotes (Figure 4c). Among the eukaryotes, fungi (Ascomycota, Basidiomycota and Glomeromycota) and Labyrinthulomycetes possessed the most connections with the prokaryotes.” (Line 354-370)

line 497-498 how was taxon detection limit determined? It needs to be explained.

To make it clearer, the sentence was changed to:

“This can be explained by the rather shallow average sequencing depth of 20,000 reads per sample, one single read would correspond to 0.05‰ in the microbiota, and thus taxa with lower abundances could not be detected.” (Line 518-521)

Conclusions section - needs to be improved to correspond with the goals set at the beginning of the article (lines 84-89)

The section has been changed to correspond with the goals (Line 549-555).

“In conclusion, this study provides microbiome-based insights into the environmental filtering effects of rewetting of fen ecosystems. Soil moisture was the driving factor for change in microbial community and GHG fluxes. In the coastal fen, the inflow of brackish water seemed to stimulate a high abundance of sulfate reducers, suppressing methanogenic activity. This suggests that, in terms of minimizing CH4 emissions after rewetting, increased sulfate content and/or top soil removal might be promising options. Future analysis of seasonal microbiome dynamics and measurements of CH4 fluxes will provide evidence, whether these predictions hold true.” (Line 569-575)

Furthermore, conclusion that increase in salinity may be a means to overcome methane emissions it a bit controversial. Increased salinity is a problem itself so authors suggest to solve a problem by creating another problem. 

We agree. Salinity was changed to “sulfate content”. Nevertheless, in coastal fens we do not see salinity as a problem.

In spite those remarks I find the paper worth publishing. 

Reviewer 3 Report

This paper contains some critical defects. My major concern with this paper is the lack of discussion with “Rewetting”. You compared several factors between “drained” and “rewetting” sites. If you want to show the “rewetting effect, you should prepare 3 plots as drained, rewetting, and constantly submerged fields. Therefore, we didn’t evaluate rewetting effects for CH4 production and pro or eukaryotic communities. You only compared wetting and drained wetlands. The effects of rewetting might disappear more than after 10 years? Moreover, I didn’t understand the novel point of this study. CH4 production and microbial community are affected by water table or salinity is well-known points. What is the original point of your study?

Why did you analyze the eukaryotic community? Their abundance was lower than prokaryotes and their contribution to GHG production is too low.

You contained plant sequence information in network analysis. However, the number of plant roots might vary in space. Does the analysis take into account the fact that a small amount of the root contamination might be considered to have a significant impact? If you analyze the relationship between plants and microorganisms, why not look at rhizosphere microorganisms?

Why the dates of CH4 emission analysis and soil sampling different? If you show the relationship between this interaction, you must collect the same day. If you showed the relationships between CH4 concentration and relative abundance of methanogen, you should show the transcriptomic analysis for methanogen because DNA still remained after they died.

Why did you analyze only mcrA gene? If you want to know about the quantity of methane-oxidizing bacteria and sulfate-reducing bacteria, you should analyze qPCR for pmoA and dsr gene. The relative abundance was not enough data because their absolute quantity is not known. 16S rRNA copy number will be one of the good indicators for it.

Author Response

Answers to reviewer 3:

This paper contains some critical defects. My major concern with this paper is the lack of discussion with “Rewetting”. You compared several factors between “drained” and “rewetting” sites. If you want to show the “rewetting effect, you should prepare 3 plots as drained, rewetting, and constantly submerged fields. Therefore, we didn’t evaluate rewetting effects for CH4 production and pro or eukaryotic communities. You only compared wetting and drained wetlands. The effects of rewetting might disappear more than after 10 years?

We disagree with the reviewer. We think that there is a misunderstanding about the definition of “rewetting” in our study. We did not intend to dissect the effect of rewetting of previously drained fens from an undisturbed, constantly wet fen. We also did not intend to monitor the short-term effect of rewetting. Rather, we focus on the long-term effects onto the microbiomes caused by rewetting of previously drained fens. Both reviewers 1 and 2 agree with the soundness of the design and the quality of the study.

To make this clearer and avoid misunderstandings of the scope of our study, we have changed the title of the study to highlight the long-term manipulation:

“Long-term Rewetting of Three Formerly Drained Peatlands Drives Congruent Compositional Changes in Pro- and Eukaryotic Soil Microbiomes Through Environmental Filtering”

Furthermore, we have made this clearer in the different sections of the manuscript, including abstract, introduction and discussion.

Moreover, I didn’t understand the novel point of this study. CH4 production and microbial community are affected by water table or salinity is well-known points. What is the original point of your study?

Our study is original in several aspects

(1) To our knowledge, this is the first field study to monitor effects of such long-term manipulations onto the microbiomes of three fen types. Since the practice of rewetting previously drained mires is gaining world-wide attention as a measure to mitigate GHG emissions from soils, our study provides key-knowledge about the effects onto the microbiomes.

(2) We agree with the reviewer that it is well known that CH4 production and microbial community are affected by water table or salinity. However, this well-known knowledge is not well characterized in peatland soils, resulting in the gap between rewetting practice and concerns regarding GHG emissions. Here, we provide the knowledge of microbial ecology behind to bridge the gap.

(3) Moreover, this is the first study to comprehensively study the pro- and eukaryotic members of peat microbiomes under impact of long-time rewetting. We highlight that rewetting has a strong effect onto the fungi, protists and small animals. Especially fungi are major direct contributors of CO2 emissions.

(4) Furthermore, the study contains investigations on three different types of peatland, which represent the three most relevant fen types in Northern Germany.

Why did you analyze the eukaryotic community? Their abundance was lower than prokaryotes and their contribution to GHG production is too low.

As stated above, eukaryotes are important members of the soil biota, significantly contributing to GHG production. For instance, especially fungi are major contributors to the emissions of CO2 through mineralization of SOC under aerobic conditions. Bacterivorous protozoa and nematodes control the bacterial communities through grazing. Our study shows that these organisms are also affected by long-term rewetting. Our study is thus not restricted to one domain of life, but monitors for the first time the system-wide effects, by analyzing several important groups of the soil biota.

You contained plant sequence information in network analysis. However, the number of plant roots might vary in space. Does the analysis take into account the fact that a small amount of the root contamination might be considered to have a significant impact? If you analyze the relationship between plants and microorganisms, why not look at rhizosphere microorganisms?

We agree with the reviewer that plant sequences should be removed from the network analysis. We have thus reanalyzed the data without plants. The new co-occurrence network shows the same topology as the previous one, see new figure 4 below.

Figure 4. Co-occurrence networks. a) Co-occurrence network of indicator ASVs associated with (A) alder carr, (C) coastal fen and (P) percolation fen, or with both alder carr and coastal fen (AC), both alder carr and percolation fen (AP) and both coastal fen and percolation fen (CP). b) Co-occurrence network of indicator ASVs associated with drained and rewetted sites. c) Co-occurrence network showing connections between prokaryotic and eukaryotic ASVs. ASVs with relative abundances lower than 0.05% were discarded. ASVs were identified as indicators associated with fen type or water condition using the indicspecies package [45], and are color-coded according to fen type and water condition (drained/rewetted) in a) and b), respectively. The relative abundances of prokaryotes and eukaryotes associated with the different fen types and water condition are shown as pie charts on the right side of a) and b), respectively. The color code in pie charts of a) and b) correspond to the color code in c). Solid lines indicate significant positive Spearman’s rank correlations (R > 0.7. P < 0.01), while dashed lines indicate significant negative correlations (R < -0.7. P < 0.01). Nodes are sized according to their degrees. (Lines 341-353)

In addition, we extensively revised the results section ‘3.2.3. Co-occurrence network analysis’ and the discussion section ‘Environmental filtering effects on pro- and eukaryotic community composition’ according to the reviewer’s suggestions (Line 327-370, see above, answer to reviewer 2).

Why the dates of CH4 emission analysis and soil sampling different? If you show the relationship between this interaction, you must collect the same day.

We agree with the reviewer that the ideal way to compare methane fluxes and methanogen abundance is to sample both at the same day. Nevertheless, methanogen abundances did not change drastically over time, especially when monitored with qPCR based on DNA. qPCR data from August 2017 show that there were no strong changes in methanogen abundance. These data were added to Table 2. Methane flux measurements of the same timepoint were not available, but were taken in July 2017. As the focus of the manuscript is on the peat soil microbial community and the environmental filtering through long-term rewetting, the GHG flux data is used to complement the findings in community data and is not shown as a central finding in this study.

Table 2. Fluxes of CH4 and ecosystem respiration (measured as CO2 flux) on all study sites compared to methanogen abundance (mcrA_DW). Negative flux values represent uptake from the atmosphere. Table shows mean values (n=5 for CH4 and ecosystem respiration CO2 fluxes, n=9 for mcrA_DW) and standard deviation. Significance was tested with Kruskal-Wallis test.

Site

CH4 flux
(mg m-2 h-1)

Ecosystem respiration (CO2) flux (mg m-2 h-1)

mcrA_DW (copies g-1 soil dry weight) April 2017

mcrA_DW (copies g-1 soil dry weight) August 2017

Alder_dry

-0.08

± 0.07

493

± 406

8.5*105

± 2.9*105

3.14*105

±3.30*104

Alder_wet

0.88*

± 0.51

47

± 70

2.1*107

± 2.6*107

3.89*107

±1.66*106

Coast_dry

-0.11

± 0.14

2024

± 585

3.5*105

± 2.5*105

3.62*105

±9.97*104

Coast_wet

0.00*

± 0.01

1420

± 311

1.8*106

± 2.1*106

5.32*105

±6.90*104

Perco_dry

-0.12

± 0.05

783

± 152

4.3*106

± 3.5*106

6.31*106

±1.07*106

Perco_wet

15.8*

± 23.8

550

± 232

3.4*107

± 1.5*107

5.28*107

±3.55*106

* significantly higher than in drained site (P < 0.05)

significantly lower than in drained site (P < 0.05)

If you showed the relationships between CH4 concentration and relative abundance of methanogen, you should show the transcriptomic analysis for methanogen because DNA still remained after they died.

We disagree. Transcriptomics, as suggested by the reviewer, would only provide relative abundance data that are difficult to compare with an ecosystem parameter, such as CH4 emission.

Why did you analyze only mcrA gene?

The usage of methanogen-specific qPCR enabled their quantification per gram soil, as opposed to relative abundance data derived from NGS amplicon data sets. Their abundance can potentially inform about methane emissions.

If you want to know about the quantity of methane-oxidizing bacteria and sulfate-reducing bacteria, you should analyze qPCR for pmoA and dsr gene.

We disagree. 16S rRNA gene amplicon data are sufficient or the comparison of the aforementioned functional groups, since we compared their relative abundance with each other. This is in fact superior to qPCR for comparative analysis, due to technical problems in quantification of functional genes and comparability of different qPCR assays.

The relative abundance was not enough data because their absolute quantity is not known. 16S rRNA copy number will be one of the good indicators for it.

We disagree, as stated in the comment above. For our analysis, the relative abundance data are sufficient, and in our opinion even superior to qPCR data.

Round 2

Reviewer 3 Report

Comments and Suggestions for Authors

Thank you for addressing the comments. I am satisfied with the comments given.

Please consider revising these. 

1) You should emphasize the 4 original points of your study in the introduction and discussion part. 

2) I disagree with your comment as "relative abundance data is enough for comparison of the functional group". Not only the relative abundance but also whole microbial biomass is also important. If possible, add DNA data as an indicator of microbial biomass

Author Response

Answer to reviewer 3

Comments and Suggestions for Authors

Thank you for addressing the comments. I am satisfied with the comments given.

Please consider revising these. 

1) You should emphasize the 4 original points of your study in the introduction and discussion part. 

As suggested by the reviewer, we highlighted the 4 original points in the manuscript more distinct. Changes in new version are underlined in the answers to reviewer.

(1) To our knowledge, this is the first field study to monitor effects of such long-term manipulations onto the microbiomes of three fen types. Since the practice of rewetting previously drained mires is gaining world-wide attention as a measure to mitigate GHG emissions from soils, our study provides key-knowledge about the effects onto the microbiomes.

This first point was already clearly emphasized in introduction (Line 80) and discussion (Line 442-445). To highlight the three different fen types, we named them again in the conclusion.

“In conclusion, this study provides microbiome-based insights into the environmental filtering effects of rewetting of three fen ecosystems, alder carr, percolation fen and coastal fen. Since the practice of rewetting previously drained mires is gaining more attention as a measure to mitigate GHG emissions from soils, our study provides key-knowledge about the effects onto the resident microbiomes.” (Line 572-576)

2) We agree with the reviewer that it is well known that CH4 production and microbial community are affected by water table or salinity. However, this well-known knowledge is not well characterized in peatland soils, resulting in the gap between rewetting practice and concerns regarding GHG emissions. Here, we provide the knowledge of microbial ecology behind to bridge the gap.

To point out the direct connection between microbiome and GHG emission we emphasized it in the introduction.

“We hypothesize that despite the obvious structural and geo genetic differences, congruent effects of rewetting will be detectable in the microbiomes of the three peatland types because of environmental filtering. Possible controlling factors of community composition, like DOM quantity and quality, soil moisture, soil/water salinity, were also determined to allow for analyzing the effect of rewetting on key players of SOM mineralization and CH4-cycling microorganisms, responsible for GHG emissions from peat soils.” (Line 84-89)

The change in microbial community and their effect on GHG emissions are already directly referred in the last abstract of the discussion (Line 562-570).

(3) Moreover, this is the first study to comprehensively study the pro- and eukaryotic members of peat microbiomes under impact of long-time rewetting. We highlight that rewetting has a strong effect onto the fungi, protists and small animals. Especially fungi are major direct contributors of CO2 emissions.

To highlight that the study on microbiomes contains prokaryotes and eukaryotes, both were named again in the introduction, discussion and conclusion.

“Our results may contribute to better understanding of how pro- and eukaryotic microbiomes respond to rewetting and to the relevant controlling environmental factors. “(Line 91-93)

“To our knowledge, this is the first comprehensive study investigating the impact of long-term rewetting (>> 10 years) on pro- and eukaryotic freshwater and coastal fen microbiomes.” (Line 442-443)

“In all systems soil moisture was the driving factor for change in pro- and eukaryotic microbial community and GHG fluxes.” (Line 576-577)

(4) Furthermore, the study contains investigations on three different types of peatland, which represent the three most relevant fen types in Northern Germany.

Three fen types and their relevance in northern Germany were pointed out:
“In this study we explore the effect of long-term rewetting on temperate fen prokaryotic and eukaryotic microbial communities. For this purpose, the three most relevant fen types in northern Germany were investigated in pairs of drained (dry) and long-term rewetted (wet) fens with contrasting water tables and vegetation: a brackish coastal fen (Coast), a percolation fen (Perco) and an alder carr (Alder).” (Line 80-84)

The three fen types were also added to conclusion.

“In conclusion, this study provides microbiome-based insights into the environmental filtering effects of rewetting of three fen ecosystems, alder carr, percolation fen and coastal fen.” (Line 572-573)

2) I disagree with your comment as "relative abundance data is enough for comparison of the functional group". Not only the relative abundance but also whole microbial biomass is also important. If possible, add DNA data as an indicator of microbial biomass

The manuscript contained already data on DNA content in supplementary table S1. A comment on DNA content has now been added to the results section. Line (268-270)

“DNA content per gram dry soil was highest in Perco and Alderwet (> 100 µg g-1 DW soil, see Table S1). In Alderwet and Percowet DNA content decreased with depth, while in other sites no distinct trend was observed.”

Round 3

Reviewer 3 Report

Dear authors

Authors responded to all my comments with enough explanation and revised adequately. It is acceptable.